# How Ultrasound Can Be Useful for Staging Chronic Kidney Disease in Dogs: Ultrasound Findings in 855 Cases

**DOI:** 10.3390/vetsci7040147

**Published:** 2020-10-01

**Authors:** Francesca Perondi, Ilaria Lippi, Veronica Marchetti, Barbara Bruno, Antonio Borrelli, Simonetta Citi

**Affiliations:** 1Department of Veterinary Science, University of Pisa, San Piero a Grado, 56122 Pisa, Italy; f.perondi87@gmail.com (F.P.); veronica.marchetti@unipi.it (V.M.); simonetta.citi@unipi.it (S.C.); 2Department of Veterinary Science, University of Turin, 10124 Turin (TO), Italy; barbara.bruno@unito.it (B.B.); antonio.borrelli@unito.it (A.B.)

**Keywords:** ultrasound, kidney, CKD, IRIS stage, dog

## Abstract

In patients affected by chronic kidney disease (CKD), some ultrasonographic (US) abnormalities have been shown to correlate better than others with the progression of the disease. The aim of the study was to evaluate the prevalence of the most frequent renal US abnormalities in dogs at different stages of CKD, and to investigate their association with CKD International Renal Interest Society (IRIS) stages. Medical records and ultrasonographical report of 855 dogs were retrospectively included. The most frequent renal ultrasonographic abnormalities were: increased cortical echogenicity, abnormal ratio of cortico-medullary junction (C/M) and pyelectasia. A statistically significant difference in the prevalence of irregular contour, abnormal cortico-medullary junction, abnormal C/M, increased cortical echogenicity, and pyelectasia was found for dogs at different IRIS stages. The number of dogs with more than one US abnormality increased significantly with the progression of IRIS stage. In conclusion, increased cortical echogenicity, abnormal C/M junction and pyelectasia were the most prevalent US abnormalities in our CKD population. Although none of the US abnormalities showed a significantly higher prevalence, the number of dogs presenting > 3 US abnormalities increased significantly from IRIS 2 to IRIS 4. Renal US is an excellent ancillary diagnostic test, which should be used together with renal functional parameters, to monitor the progression of CKD.

## 1. Introduction

Chronic kidney disease (CKD) is the most commonly recognized renal disease in dogs, and it is defined as any structural and/or functional abnormality of one or both kidneys, which has been continuously present for at least three months [1,2]. The International Renal Interest Society (IRIS) stages of CKD are based on the evaluation of serum creatinine, symmetric dimethylarginine (SDMA), proteinuria, and blood pressure, in order to facilitate diagnosis, prognostic evaluation, and treatment of the disease [1,3]. Abdominal ultrasonography (US) is routinely performed and recommended in animals with renal disease, or when creatinine serum levels are increased [4,5,6]. Ultrasonography is the first imaging investigation required in both human and veterinary patients with chronic renal failure [4,7]. The ultrasound characterization of chronic nephropathy is based on subjective criteria, such as cortical echogenicity, kidney shape, kidney size and internal architecture [4]. Increased cortical echogenicity and decreased corticomedullary differentiation are reported to be the more common signs of CKD in veterinary medicine [4,6]. These changes are suggestive of chronic and irreversible lesions of the kidney; however, both dogs and cats may have CKD without showing renal changes at ultrasonography [6,8]. In human medicine, simultaneous assessment of cortical echogenicity, kidney size and serum creatinine levels were evaluated to determine the chronicity of the disease [9]. In human CKD patients, renal echogenicity correlates better with serum creatinine, than other ultrasound parameters (longitudinal size, parenchymal thickness, and cortical thickness) and it seems a good parameter to estimate renal function [7]. 

The aim of the present study was to evaluate the prevalence of the most frequent renal US abnormalities in dogs at different stages of CKD, and to investigate their association with CKD IRIS stages.

## 2. Materials and Methods 

Medical records of all dogs with diagnosis of CKD, which presented to the Veterinary Teaching Hospital “Mario Modenato” of Pisa University, between January 2013 and December 2019 were retrospectively evaluated. Inclusion criteria included documented history of CKD, laboratory findings consistent with CKD and ultrasonographic examination. Diagnosis of CKD based on chronic history (more than 3–4 months) of azotemia, polyuria-polydipsia, and inappropriate urine-specific gravity. Enrolled dogs may or may not be affected by proteinuria and/or systemic hypertension. Exclusion criteria included historical, laboratory and/or ultrasonographic findings consistent with acute kidney injury (AKI). For each dog included in the study, data regarding history, physical examination, biochemical profile (serum urea, creatinine, total calcium, phosphate, calcium-phosphate product) and US report were collected from the medical record. Dogs were classified according to the IRIS guidelines for CKD on the basis of serum creatinine concentration as it follows: stage 2, 1.4 to 2.0 mg/dL; stage 3, 2.0 to 5.0 mg/dL; and stage 4, >5.0 mg/dL [3].

### 2.1. Ultrasonographic Procedures 

Aplio 400 Toshiba^®^ ultrasound system (Toshiba, Italy), with a microconvex (5–7.5 MHz) and linear (7–12 MHz) transducer (6–8 MHz) was used by two radiologists with a 20-year experience in performing ultrasound of the kidneys.

For each dog of the study, US reports and recorded images were reviewed for the following US findings [6,8,10].

Renal Contour: regular or irregular contour.Cortico-medullary junction (C/M junction): normal or abnormal. C/M junction was considered abnormal for shaded or absent, enhanced or hyperechoic differentiation.Cortical and medullary ratio (C/M ratio): Normal (C/M ratio > 1:1) or abnormal (C/M ratio < 1:1). Cortical thickness was measured in the sagittal plane over a medullary pyramid, perpendicular to the capsule. Medullary thickness was measured from the renal hilum to the outer margin of the cortico-medullary junction (Figure 1).Cortical echogenicity: normal or abnormal (increased cortical echogenicity compared to the echogenicity of the liver and spleen).Medullary echogenicity: normal or abnormal (increased medullary echogenicity compared to the echogenicity of the renal cortex).Cysts: presence or absence of cortical cysts that appear oval shaped, with anechoic content.Mineralization: presence or absence. Renal mineralization was defined as dispersed hyperechoic foci.Infarcts: presence or absence of renal infarcts. Infarcts were defined as wedge-shaped hyperechoic areas, with a broad base at the surface of the kidney, that narrows toward the corticomedullary junction, resulting in localized thinning of the cortex and renal contour defect.Pyelectasia: presence or absence. Pyelectasia was defined as a renal pelvis dilation > 3–4 mm.Peri-renal effusion: presence or absence of peri-renal fluid.

### 2.2. Statistical Analysis

Statistical analysis was performed by the use of a commercial statistical software (Graphpad prism, 6 for Mac). Continuous variables were tested for normality through the Kolmogorov-Smirnov test. Descriptive statistics included computation of means and standard deviation for normally distributed variables such as creatinine, urea, calcium, phosphate and serum calcium-phosphorus concentration product (sCaPP) and medians with minimum and maximum ranges for non-normally distributed variables (age). 

Chi-squared test was used to compare the prevalence of US abnormalities at different CKD IRIS stages. 

Chi squared test was used to compare the prevalence of one, two to three, or more than three US abnormalities in different CKD IRIS stages.

Kruskall Wallis test was used to compare the prevalence of elevated (>70 mg^2^/dL^2^) serum calcium-phosphate product (sCaPP) in dogs showing kidney mineralization, at different CKD IRIS stages. All data were considered statistically significant for *p* value < 0.05. 

## 3. Results

Retrospective medical record evaluation of our electronic clinical database identified 865 dogs affected by CKD. Ten dogs with clinical, ultrasound and laboratory signs of CKD IRIS stage 1 were excluded from statistical analysis, due to the low number (*n* = 10). A total of 855 dogs were included in the present study. Of the 855 enrolled dogs, 337 dogs (39.40%) were in CKD IRIS stage 2, 295 dogs (34.50%) were in IRIS stage 3 and 223 dogs (26%) were in IRIS stage 4. According to gender, 430 dogs (50.30%) were intact males, 35 dogs (4%) were castrated males, 240 dogs (28.20%) were intact females, and 150 dogs (17.50%) were spayed females. Median age was 9 years (range 0.40–17 years). The most represented breeds were mixed-breed (*n* = 258; 30%), Labrador (*n* = 70; 8%), German Shepherd (*n* = 46; 5.30%), Boxer (*n* = 34; 3.90%) and Golden Retriever (*n* = 31; 3.50%). The mean values of serum creatinine, urea, calcium, phosphorus and sCaPP for each IRIS group were reported in Table 1.

The Chi squared test showed a significantly different prevalence of US findings of irregular contour, abnormal C/M junction, abnormal C/M ratio, increased cortical echogenicity and pyelectasia for dogs at different CKD IRIS stages. No statistically significant difference was found in the prevalence of cysts, mineralization, infarcts, and peri-renal effusion (Table 2). 

For 6/10 ultrasound abnormalities (irregular contour, abnormal C/M junction, abnormal C/M ratio, increased cortical echogenicity, abnormal medullary echogenicity and pyelectasia), a significant increase in their prevalence was found with the progression of the CKD IRIS stage (Table 2). In the different CKD IRIS stages, the most prevalent US abnormalities were increased cortical echogenicity, abnormal C/M junction and pyelectasia (Figure 2).

Chi square test showed a statistically significant difference (*p* = 0.0001) in the prevalence of dogs with one, two to three, or more than three US abnormalities at different CKD IRIS stages. The number of dogs with more than one US abnormalities increased significantly with the progression of CKD IRIS stage (Table 3). 

The number of dogs with sCaPP > 70 mg^2^/dL^2^ increased with the progression of the IRIS stage (*p* = 0.0001). However, the number of dogs showing kidney mineralization among dogs with sCaPP > 70 mg^2^/dL^2^ did not differ significantly in the different CKD IRIS stages (Table 4). 

## 4. Discussion

In our study, the number of US abnormalities increased significantly with the progression of IRIS stages; in IRIS stage 4 the 60.5% of dogs presented more than three renal abnormalities, while the 39.5% of dogs presented two or three abnormalities. The main US abnormalities reported in dogs and cats with CKD were increased echogenicity, irregular contour, decrease or absence of cortico-medullary differentiation and decreased renal volume. These findings were more common in advanced stages of CKD, and were considered indicative of an irreversible condition and poor prognosis [6]. 

In our study, increased cortical echogenicity was the most common abnormality in CKD dogs, and its prevalence increased significantly with the progression of the IRIS stage (47% of dogs in IRIS 2, 69% of IRIS 3 and 91% of IRIS 4). Renal cortical echogenicity is defined by comparison with other abdominal organs, and it should be slightly hypoechoic or isoechoic to the liver, and hypoechoic to the spleen [6]. In physiological conditions, renal cortex and medulla must be clearly differentiated from each other, with a hyperechoic cortex compared to medulla [6]. Increased cortical echogenicity is the most frequent ultrasound abnormality in dogs with CKD (88%), which may be indicative of fibrosis, sclerosis or infiltrations [6,11]. In veterinary medicine, increased cortical echogenicity may be reported in cases of glomerulonephritis, amyloidosis, interstitial nephritis, acute tubular necrosis or nephrosis, end stage renal disease (ESRD) and nephrocalcinosis [6]. In particular, in a recent study in dogs, increased cortical echogenicity was correlated with glomerulosclerosis and fibrosis, but not with inflammatory lesions, such as interstitial nephritis, acute tubular necrosis and tubular atrophy [4]. In human studies, the increase in renal echogenicity was also correlated with serum creatinine levels [7], and glomerular filtration rate (GFR) [12,13]. The degree of renal echogenicity presented a significant positive correlation with serum creatinine and with CKD stage; this correlation was greater than other ultrasound changes considered in the study, such as longitudinal size, parenchymal and cortical thickness [7]. In veterinary medicine, only one study compared renal ultrasound changes with serum creatinine [14], while a recent prospective study compared US abnormalities with GFR calculated by scintigraphy [15]. In the first study, the majority of dogs presented increased diffuse renal cortical echogenicity. Increased cortical echogenicity was also present in dogs with serum creatinine within the reference range, showing that this change may occur in the early stages of the kidney disease in dogs [14]. In the second study, abnormal kidney shape, cortical hyperechogenicity, medullary hyperechogenicity and low kidney volume were found associated with low GFR estimated by scintigraphy normalized to plasma volume [15].

In our population, abnormal C/M junction (Figure 1) was the second most frequent US disorder, with a prevalence of 43% in CKD IRIS 2, 59% in CKD IRIS 3 and 69% in CKD IRIS 4 (Figure 2). This prevalence increased statistically with the progression of the IRIS stage, as reported in Table 2. Inflammatory renal diseases (interstitial nephritis, glomerulonephritis and pyelonephritis) may be often associated with a decrease in cortico-medullary distinction [5]. Decreased cortico-medullary distinction has been described in 54% of dogs with CKD, with absence of differentiation in 35% [11]. No data on cortico-medulary distinction in dogs at different stages of CKD are present. On the other hand, specific ultrasound abnormalities have been described for cats at different stages of CKD. Mild changes in cortico-medullary differentiation were reported in 25% of cats in CKD IRIS 1, and in 41.2% of cats in CKD IRIS 2. More marked changes in cortico-medullary distinction were present in 75% of cats in CKD IRIS 3 and 4 [6]. 

Although renal pyelectasia has not been considered as a common finding in dogs affected by CKD by a recent review [6], in our cohort of dogs it was the third most prevalent US disorder. Pyelectasia was presented in 25% of dogs in CKD IRIS 2, 44% of CKD IRIS 3 and 50% of CKD IRIS 4. A larger degree of renal pelvis dilation would be expected for pyelonephritis than for other inflammatory renal diseases [5]. As ureter dilatation was present in none of our patients with pyelectasia, the presence of pyelonephritis was considered the most plausible cause. Pyelonephritis may be present in some veterinary patients as a complication of CKD [1]. One study found ultrasonography to be 82% sensitive and 100% specific for the detection of mild to moderate pyelonephritis [16]. No data regarding the prevalence of pyelectasia in dogs with CKD were actually reported in veterinary medicine. In a recent retrospective study, pyelonephritis was diagnosed in the 40% of dogs with CKD, with a combination of US findings and positive urine culture [17]. Alteration of renal contour, echostructure, cortical medullary ratio and medullary echogenicity seemed to increase significantly with progression of CKD. US evidence of irregular profile (or contour) is a common abnormality in advanced stages of CKD, which may be associated with renal fibrosis [18]. Irregular profile has been commonly recognized in chronic pyelonephritis, advanced stages of glomerulonephritis, renal dysplasia, and ESRD [6]. Notomi et al. observed irregular renal profile in 68% of dogs with CKD [11], while Koch et al. found irregular renal profile in 88.3% of dogs with CKD, in association with other US abnormalities, such as a diffuse increase in echogenicity and loss of cortico-medullary ratio and asymmetry [14].

In our study, the prevalence of renal cysts, mineralization, renal infarcts and peri-renal effusion did not increase significantly with the progression of the IRIS stage. The US finding of renal cysts and infarcts may be incidental or associated with degenerative CKD. Particularly, chronic renal infarcts may be a common finding in older animals [5]. Kidney cysts are often located within the renal cortex or at the C/M junction. The kidney cysts are often associated with abnormalities of renal cortex and C/M junction. The increase in size and number of cysts can be associated not only with an alteration of the renal profile, but also with a reduction in the renal function [5]. In our cohort of dogs, no association between the presence of renal mineralization and the progression of CKD was found, despite sCaPP increased statistically in dogs with severe renal disease. Renal mineralization may be observed as a distinctive, white gritty band, located adjacent to the cortical medullary junction [19,20]. Calcification of renal parenchyma is usually mild, and randomly distributed, affecting primarily tubular epithelium, and it is often associated with chronic or severe hypercalcemia [19]. Nephrocalcinosis may be seen in CKD dogs, especially in association with conditions of hypercalcemia, hyper-vitaminosis D and nephrotoxicity [10]. Nephrocalcinosis may be concurrently associated with fibrosis and infiltration of interstitial tissue with mononuclear cells [19]. In CKD dogs, elevated sCaPP product has been associated with an increased risk of mortality [21,22]. Metastatic calcifications have been suggested to develop when sCaPP is >70 mg^2^/dL^2^ and high sCaPP levels correlated with progression of the disease [21,23,24]. In our study, sCaPP increased with progression of the IRIS stage, as previously found by Lippi et al. [21]. However, no correlation between elevation in sCaPP and increased prevalence of renal mineralization was found. The lack of correlation between elevated sCaPP and increased risk of renal mineralization was unexpected. In humans, CKD is associated with cardiovascular disease, in particular atherosclerosis and vascular calcification, mineral and bone disorder, fractures and high risk of mortality (especially cardiovascular mortality) [23,25,26,27]. In human patients, high levels of sCaPP and high phosphate levels may lead to secondary hyperparathyroidism and are associated with calcification of softs tissues [28]. It is possible that in our cases, high levels of sCaPP may be associated with mineralization in other sites than kidneys (such as stomach, myocardium, lung and liver), as reported in patient with CKD [29,30]. 

The study presented several limitations: first of all, because of the retrospective nature of the study, it was not possible to enroll the same number of dogs for each IRIS stage. For the same reason, dogs in CKD IRIS 1 (with normal serum creatinine), were excluded from the study, due to the low number of dogs compared to the other groups. Unfortunately, GFR or SDMA were present only in a limited number of patients, thus, they were not included in the statistical analysis. Further studies, including those focusing on CKD IRIS 1 dogs, are warranted. 

## 5. Conclusions

In conclusion, increased cortical echogenicity, abnormal C/M junction and pyelectasia were the most prevalent US abnormalities in our CKD population. Although none of the renal US abnormalities showed a significantly higher prevalence over the others in different IRIS stages, the number of dogs presenting > 3 US abnormalities increased significantly from IRIS 2 to IRIS 4. Renal US is an excellent ancillary diagnostic test, which should be used together with renal functional parameters, to monitor the progression of CKD over time. 

## Figures and Tables

**Figure 1 vetsci-07-00147-f001:**
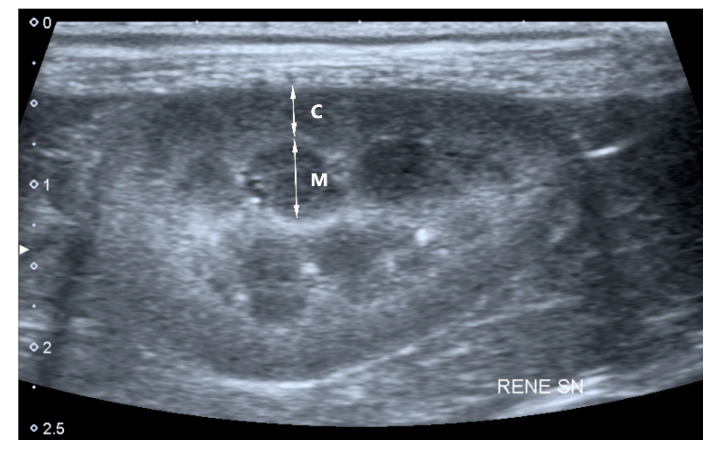
Sagittal plane of the left kidney of a 10-year-old dog with chronic kidney disease for the measurement of the cortical and medullary ratio (C/M ratio): cortical thickness (C) was measured over a medullary pyramid, perpendicular to the capsule; medullary thickness (M) was measured from the renal hilum to the outer margin of the cortico-medullary junction. In this dog the C/M ratio was 0.64.

**Figure 2 vetsci-07-00147-f002:**
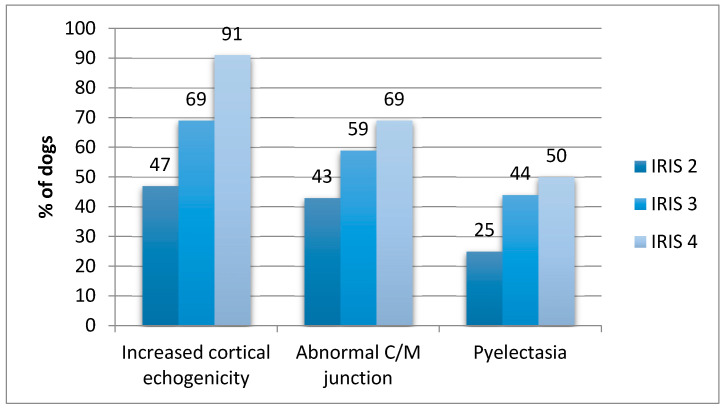
Most prevalent US abnormalities at different CKD IRIS stages.

**Table 1 vetsci-07-00147-t001:** Mean values and standard deviation (SD) of serum creatinine (mg/dL), urea (mg/dL), calcium (mg/dL), phosphate (mg/dL) and serum calcium-phosphorus concentration product (sCaPP) product (mg^2^/dL^2^) for each International Renal Interest Society (IRIS) stage.

Parameter	IRIS 2	IRIS 3	IRIS 4	Reference Range
Creatinine	1.59 ± 0.17	3.10 ± 0.80	9.40 ± 3.80	0.60–1.50
Urea	71.80 ± 42	152.70 ± 78.90	328.10 ± 123.80	15–55
Calcium	10.30 ± 1.50	10.60 ± 1.80	10.50 ± 2.30	8.7–11.8
Phosphuros	4.70 ± 1.60	6.60 ± 3.10	14.20 ± 5.80	2.5–5.0
sCaPP	48.20 ± 15.80	69.70 ± 31.10	143.20 ± 55.40	<70

**Table 2 vetsci-07-00147-t002:** Chi-squared test of the prevalence of ultrasonographic (US) abnormalities at different chronic kidney disease (CKD) IRIS stages.

Parameter	IRIS 2	IRIS 3	IRIS 4	*p* Value
Irregular contour	33/337 (9.8%)	52/295 (17.6%)	54/223 (24.2%)	0.0185 *
Abnormal C/M junction	144/337 (42.7%)	174/295 (59%)	153/223 (68.6%)	0.0035 *
Abnormal C/M ratio	30/337 (89%)	68/295 (23%)	66/223 (29.5%)	0.0049 *
Increased cortical echogenicity	158/337 (46.9%)	203/295 (68.8%)	203/223 (91%)	0.0001 *
Abnormal medullary echogenicity	59/337 (17.5%)	90/295 (30.5%)	95/223 (42.6%)	0.0018 *
Pyelectasia	85/337 (25.2%)	131/295 (44.4%)	111/223 (49.7%)	0.0018 *
Cysts	51/337 (15.1%)	58/295 (19.7%)	58/223 (26%)	0.2096
Mineralization	59/337 (17.5%)	74/295 (25.1%)	54/223 (24.2)	0.4688
Infarcts	10/337 (3%)	9/295 (3%)	6/223 (2.7%)	0.9999
Peri-renal effusion	4/337 (1.2%)	9/295 (3%)	14/223 (6.3%)	0.1401

C/M junction: cortico-medullary junction; C/M ratio: cortico-medullary ratio. * *p* value < 0.05.

**Table 3 vetsci-07-00147-t003:** Prevalence of one or more ultrasound abnormalities in different CKD IRIS stages.

Number of US Abnormalities	IRIS 2	IRIS 3	IRIS 4	*p* Value
1	128/337 (38%)	42/295 (14.2%)	0/223 (0%)	0.0001 *
2–3	129/337 (38.3%)	123/295 (41.7%)	88/223 (39.5%)	
>3	80/337 (23.7%)	130/295 (44.1%)	135/223 (60.5%)	

* *p* value < 0.05.

**Table 4 vetsci-07-00147-t004:** Kruskall Wallis test of the prevalence of elevated sCaPP (>70 mg^2^/dL^2^) in dogs showing kidney mineralization, at different CKD IRIS stages.

Number of US Abnormalities	IRIS 2	IRIS 3	IRIS 4	*p* Value
sCaPP > 70	38/212 (17.9%)	124/218 (56.7%)	198/206 (96.1%)	0.0001 *
Mineralization	8/38 (21%)	30/124 (24.2%)	48/198 (24.2%)	0.8442

* *p* value < 0.05.

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
