# Peer review of "How Ultrasound Can Be Useful for Staging Chronic Kidney Disease in Dogs: Ultrasound Findings in 855 Cases"

_vetsci, 2020, doi:10.3390/vetsci7040147_

Round 1

Reviewer 1 Report

Dear authors,
This is an interesting study that describe the prevalence of the most frequent renal US abnormalities in dogs at different stages of CKD, and  investigate their association with CKD IRIS stages.

My specific comments are listed below:

ABSTRACT:

Line 20: I suggest spelling US as ultrasonographic and putting in between parenthesis “(US)”

Line 22: I would suggest putting between parenthesis (C/M) after "cortico-medullary junction" and adding "ratio" to  "abnormal C/M"

INTRODUCTION: 

Line 36: I suggest add "it is" after "staging of CKD"

Line 40: I would change "ultrasound" by "sonographic"

Line 42: I would add the word "renal or kidney" before shape and size

MATERIAL AND METHODS

Line 79: It would be good and clarifying to add a figure of an ultrasonographic image of a kidney in sagittal plane where calipers or arrowheads could show how to measure cortical and medullary thickness to calculate the C/M ratio

Line 84: I would add "shaped" after oval.

RESULTS:

Line 114: about the mixed-breed dogs which were the most represented breed n=258 (30%), it would be of interest to know how many of them weighed less than 10 kg, how much weighed between 10 to 20 kg, and how many weighed more than 20 kg. Taking into account that the rest of the breeds belonged to the group of large breeds would be interesting to know if perhaps there is a higher prevalence in dogs that weigh more than for example 20 kg..

Line 119: Table 1. Please delete the sentence (legend) as it correspond to Table 2. Anyway, I understand you have not found significant differences between the values of the different parameters for each IRIS group, but it should be comment in the text as there are parameters as the creatinine that shows a value of 1.59 ± 0.17 in IRIS 2 and increase its value to 9.40± 3.80 in IRIS 4, and similarly occurs with the Phosphurus, sCaPP and urea. If the standard deviation are to big in some cases maybe there are not statistically significant differences but I do think it should be mentioned or analyzed in the text. 

Lines 133-134: I would suggest rephrasing this sentence "Chi square test showed a statistically significant difference (p=0.0001) in the prevalence of dogs with one, 2 to 3, or more than 3 US abnormalities at different CKD IRIS stages" as is a bit confusing.. do you mean there were statistically significant differences in all the IRIS stages comparing between them in each group of US abnormalities? as in the group of 2-3 abnormalities does not seem to be significant differences between the IRIS stages.. or are you comparing the data in other way? Perhaps you should add in every value of the table a letter (a, b or c) (superscript) and as a legend bellow the table: a,b,cDifferent letter indicate significant differences (p < 0.05) or no letters but * (asteric symbol) and: * In the same row indicate significant differences (p<0.05).

Line 137: Table 3: there are two mistakes, you should change 339 for 337 (>3 number of US abnormalities, IRIS 2). Add the symbol % to the value 88/223 (39.5) (2-3 abnormalities, IRIS 4)

DISCUSSION

Line 168: Please change "then" for "than".

Line 200: Please add the year of publication of your reference: Notomi and Colleagues (2006). Furthermore, not sure if is correct the word "colleagues" or just add the first author "et al.". Please revise the instructions for authors at the web site 

Line 201: If necessary change "Colleagues" by " et al."

Line 215 and 217: The term hypercalcemia should be written the same way (not hyper-calcemia).

Lines 218-221: I would suggest a reference after the sentence "In CKD dogs, elevated... ... correlated with progression of the disease", unless the bibliographic reference for this sentences is also Lippi et al.

Line 222: Change if necessary "Lippi and Colleagues" by "Lippi et al" and add the date of the publication (2014)

Line 229: You affirm that high levels of sCaPP may be associated with mineralization in other sites.. could you put any example? and give a reference where you have found that this mineralization can occurs?

Line 231: I suggest to delete the word "also"

REFERENCES

Line 266: Reference 6: There is newer editions of this book, it would be advisable to put the most recent edition (3rd ed in 2015).

Line 281: There is a strange symbol in the tittle of the reference, between the words "urinary" and "tract" maybe is an informatics error.

Line 286. Reference 15: the abbreviate name of the journal is not written in cursive.. please revise the instructions for authors at the web site and unifie all the references.

Author Response

Dear authors,
This is an interesting study that describe the prevalence of the most frequent renal US abnormalities in dogs at different stages of CKD, and  investigate their association with CKD IRIS stages.

 Thank you for your comments and revison

My specific comments are listed below:

ABSTRACT:

Line 20: I suggest spelling US as ultrasonographic and putting in between parenthesis “(US)”

- Done

Line 22: I would suggest putting between parenthesis (C/M) after "cortico-medullary junction" and adding "ratio" to  "abnormal C/M"

- Done

INTRODUCTION: 

Line 36: I suggest add "it is" after "staging of CKD"

 -Done

Line 40: I would change "ultrasound" by "sonographic"

- Done

Line 42: I would add the word "renal or kidney" before shape and size

- done

MATERIAL AND METHODS

Line 79: It would be good and clarifying to add a figure of an ultrasonographic image of a kidney in sagittal plane where calipers or arrowheads could show how to measure cortical and medullary thickness to calculate the C/M ratio

- In agreement with Reviewer, an ultrasonographic image of a kidney in sagittal plane has been included in the MM section. Please see figure 1.

Line 84: I would add "shaped" after oval.

 - Done

RESULTS:

Line 114: about the mixed-breed dogs which were the most represented breed n=258 (30%), it would be of interest to know how many of them weighed less than 10 kg, how much weighed between 10 to 20 kg, and how many weighed more than 20 kg. Taking into account that the rest of the breeds belonged to the group of large breeds would be interesting to know if perhaps there is a higher prevalence in dogs that weigh more than for example 20 kg..

Thank you for the comment. Unfortunately the weight was not included between parameters analyzed and now it is very difficult to recover this data.

Line 119: Table 1. Please delete the sentence (legend) as it correspond to Table 2. Anyway, I understand you have not found significant differences between the values of the different parameters for each IRIS group, but it should be comment in the text as there are parameters as the creatinine that shows a value of 1.59 ± 0.17 in IRIS 2 and increase its value to 9.40± 3.80 in IRIS 4, and similarly occurs with the Phosphurus, sCaPP and urea. If the standard deviation are to big in some cases maybe there are not statistically significant differences but I do think it should be mentioned or analyzed in the text. 

- Thank you for your comment. The legend’s sentence has been removed: it was a mistake. About table 1, any statistical analysis has been done in this case , but we opted to report only means values of the different parameters for each iris stages.

Lines 133-134: I would suggest rephrasing this sentence "Chi square test showed a statistically significant difference (p=0.0001) in the prevalence of dogs with one, 2 to 3, or more than 3 US abnormalities at different CKD IRIS stages" as is a bit confusing.. do you mean there were statistically significant differences in all the IRIS stages comparing between them in each group of US abnormalities? as in the group of 2-3 abnormalities does not seem to be significant differences between the IRIS stages.. or are you comparing the data in other way? Perhaps you should add in every value of the table a letter (a, b or c) (superscript) and as a legend bellow the table: a,b,cDifferent letter indicate significant differences (p < 0.05) or no letters but * (asteric symbol) and: * In the same row indicate significant differences (p<0.05).

- Thank you for the comment. Chi square test showed a significant difference but with this analysis we could not showed in detail the different between each iris group of US abnormaities..for this reason we could not added different letter to indicate the specific signicant difference. If you prefer, we could put in the manuscript the figure of this test.

Line 137: Table 3: there are two mistakes, you should change 339 for 337 (>3 number of US abnormalities, IRIS 2). Add the symbol % to the value 88/223 (39.5) (2-3 abnormalities, IRIS 4)

- Done

DISCUSSION

Line 168: Please change "then" for "than".

-Done

Line 200: Please add the year of publication of your reference: Notomi and Colleagues (2006). Furthermore, not sure if is correct the word "colleagues" or just add the first author "et al.". Please revise the instructions for authors at the web site 

- Done

Line 201: If necessary change "Colleagues" by " et al."

- Done

Line 215 and 217: The term hypercalcemia should be written the same way (not hyper-calcemia).

- Done

Lines 218-221: I would suggest a reference after the sentence "In CKD dogs, elevated... ... correlated with progression of the disease", unless the bibliographic reference for this sentences is also Lippi et al.

- Done

Line 222: Change if necessary "Lippi and Colleagues" by "Lippi et al" and add the date of the publication (2014)

- Done

Line 229: You affirm that high levels of sCaPP may be associated with mineralization in other sites.. could you put any example? and give a reference where you have found that this mineralization can occurs?

In agreement with reviewer’s comment the sentence has been changed as it follows: “It is possible that in our cases, high levels of sCaPP may be associated with mineralization in other sites rather than kidneys (such as stomach, myocardium, lung and liver) as reported in patient with CKD” and the new references has been included in the manuscript.

Line 231: I suggest to delete the word "also"

- Done

REFERENCES

Line 266: Reference 6: There is newer editions of this book, it would be advisable to put the most recent edition (3rd ed in 2015).

- Done

Line 281: There is a strange symbol in the tittle of the reference, between the words "urinary" and "tract" maybe is an informatics error.

- Done

Line 286. Reference 15: the abbreviate name of the journal is not written in cursive.. please revise the instructions for authors at the web site and unifie all the references.

- Done

Reviewer 2 Report

line 119: (Table 1) It should be deleted, it is not necessary for table 1, but only for Table 2

line 148: 39.50in dogs% ...change in 39.50%

______________________________________________

The manuscript outline how ultrasonography can be useful in staging chronic renal disease in dogs, presenting a retrospective study on patients collected at the Veterinary teaching Hospital of Pisa University, Italy.

The paper is well organized, with proper statistical analysis, and well presented. Despite the fact that the novelty of the treated topics is low, must be considered the number of dogs considered (855) which gives importance to the results and to the discussion presented. In any case the manuscript gives some indications that might be helpful to the reader.

Author Response

line 119: (Table 1) It should be deleted, it is not necessary for table 1, but only for Table 2

In agreement with Reviewer comment the legend’s sentence has been removed form the table 1.

line 148: 39.50in dogs% ...change in 39.50%

- done

______________________________________________

The manuscript outline how ultrasonography can be useful in staging chronic renal disease in dogs, presenting a retrospective study on patients collected at the Veterinary teaching Hospital of Pisa University, Italy.

The paper is well organized, with proper statistical analysis, and well presented. Despite the fact that the novelty of the treated topics is low, must be considered the number of dogs considered (855) which gives importance to the results and to the discussion presented. In any case the manuscript gives some indications that might be helpful to the reader.

Thank you for your comment and revision.